# BASIDIN as a New Protein Effector of the Phytopathogen Causing Witche’s Broom Disease in Cocoa

**DOI:** 10.3390/ijms241411714

**Published:** 2023-07-20

**Authors:** Keilane Silva Farias, Monaliza Macêdo Ferreira, Geiseane Veloso Amaral, Maria Zugaib, Ariana Silva Santos, Fábio Pinto Gomes, Rachel Passos Rezende, Karina Peres Gramacho, Eric Roberto Guimarães Rocha Aguiar, Carlos Priminho Pirovani

**Affiliations:** 1Centro de Biotecnologia e Genética, Departamento de Ciências Biológicas, Universidade Estadual de Santa Cruz, Rodovia Ilhéus–Itabuna, km 16, Ilhéus 45662-900, Bahia, Brazil; keilaneuesc@gmail.com (K.S.F.); monalizamacedo2@gmail.com (M.M.F.); geiseane_veloso@hotmail.com (G.V.A.); mariazugaib@hotmail.com (M.Z.); ana.silva0491@gmail.com (A.S.S.); rachel@uesc.br (R.P.R.); ergraguiar@uesc.br (E.R.G.R.A.); 2Fisiologia Vegetal, Departamento de Ciências Biológicas, Universidade Estadual de Santa Cruz, Rodovia Ilhéus-Itabuna, km 16, Ilhéus 45662-900, Bahia, Brazil; gomes@uesc.br; 3Comissão Executiva do Plano da Lavoura Cacaueira, Centro de Pesquisas do Cacau–MAPA, Laboratório de Fitopatologia Molecular, km 22 Rodovia Ilhéus Itabuna, Ilhéus 45600-970, Bahia, Brazil

**Keywords:** effectors, basidiomycetes, witche’s broom, hypersensitivity response, *Theobroma cacao*

## Abstract

The fungus *Moniliophthora perniciosa* secretes protein effectors that manipulate the physiology of the host plant, but few effectors of this fungus have had their functions confirmed. We performed functional characterization of a promising candidate effector of *M. perniciosa*. The inoculation of rBASIDIN at 4 µmol L^−1^ in the mesophyll of leaflets of *Solanum lycopersicum* caused symptoms of shriveling within 6 h without the presence of necrosis. However, when sprayed on the plant at a concentration of 11 µmol L^−1^, it caused wilting symptoms only 2 h after application, followed by necrosis and cell death at 48 h. rBASIDIN applied to *Theobroma cacao* leaves at the same concentration caused milder symptoms. rBASIDIN caused hydrogen peroxide production in leaf tissue, damaging the leaf membrane and negatively affecting the photosynthetic rate of *Solanum lycopersicum* plants. Phylogenetic analysis indicated that BASIDIN has orthologs in other phytopathogenic basidiomycetes. Analysis of the transcripts revealed that BASIDIN and its orthologs are expressed in different fungal species, suggesting that this protein is differentially regulated in these basidiomycetes. Therefore, the results of applying BASIDIN allow the inference that it is an effector of the fungus *M. perniciosa*, with a strong potential to interfere in the defense system of the host plant.

## 1. Introduction

The fungus *Moniliophthora perniciosa* has great genetic variability [1] and can colonize and infect plants of the families *Malvaceae*, *Solanaceae*, *Bignoniaceae*, *Bixacea* and *Malpighiacea* [1,2,3]. This fungus causes witche’s broom disease of the cocoa tree, which is responsible for major crop losses throughout the Americas [4].

The control of the disease is accomplished through phytosanitary pruning, application of fungicides, biological control and/or use of resistant varieties. Containing the disease has become difficult since the climatic conditions of intermittent periods of rain and drought favor the life cycle of the fungus and its proliferation [5].

For *M. perniciosa* to colonize the cacao tree, it expresses a range of genes to modulate the host’s immune response and thus reach the final stage of the disease [6]. Within this context, studies of this plant–pathogen interaction are necessary to improve understanding of the invasion and virulence mechanisms of the fungus and thus develop disease control practices that are efficient and result in increased economic viability of crops.

The repertoire of secreted effector proteins was predicted from the genome sequencing of five isolates from subpopulations of *M. perniciosa* (three from cacao and two from Solanaceae), along with one isolate of *Moniliophthora roreri* from Peru [7]. Knowledge of the functions of effector proteins can contribute to the unraveling of the mechanism of pathogen-host interaction. Thus, the functional characterization of these candidate effectors is important to confirm the involvement of these proteins in the pathogenesis of *M. perniciosa*.

Effectors are considered crucial for disease establishment and progression since they are expressed by pathogens to modulate physiology or interfere with the plant’s innate immunity, promoting virulence or activating the host’s defense system [8,9]. This activation is based on recognition of the effector Avr by host resistance (R) proteins, which leads to a signaling cascade, with hypersensitivity response (HR), synthesis of secondary metabolites, production of reactive oxygen species, and localized death of infected cells, thus limiting the progress of the pathogen [8,10].

These effectors can act at different locations in the host cells, performing various functions, such as acting as toxins [9], suppressing PTI responses (immunity triggered by PAMPs) [10], degrading cell wall enzymes [11], and inhibiting proteases [12], among others. Thus, studies of these effectors can enable the discovery of strategies to develop plants resistant to pathogen attacks.

Despite great efforts to identify and characterize fungal effector proteins, many secreted proteins with short, cysteine-rich sequences and without conserved domains that can be effectors are still considered hypothetical, since they have no homologs in the databases with their respective characterized functions. A particular difficulty in characterizing fungal effectors is the lack of significant conservation of their structures since they do not share conserved domains among themselves that provide indications about their possible functions [8]. However, studies of effectors have become very important for understanding pathogenicity, resistance, host specificity, and how silencing effector genes can interfere with the virulence mechanism of the pathogen, impairing its ability to invade host tissues [13,14].

In this context, to contribute to the understanding of witche’s broom disease of the cacao tree caused by the fungus *M. perniciosa*, we characterized a putative effector protein, selected from our genomic bank [7] under the code Mp4145-3305, which has 100% identity with the predicted protein (GenBank: EEB89748.1) in the first genomic sequencing of the fungus *M. perniciosa* [15]. This protein was expressed in a heterologous system and the recombinant protein showed typical effector activity in a bioassay with *T. cacao* and model tomato plants (*Solanum lycopersicum*, var. Micro-Tom—Sol Genomic network SL4.0 and Annotation ITAG4.0). It was named BASIDIN, and is restricted to the basidiomycetes group.

## 2. Results

### 2.1. In Silico Selection, Expression, and Purification of the Recombinant Effector Candidate

The BASIDIN coding sequence is 486 nucleotides (nt) in length and has 162 amino acid residues (aa). Analyses using the SignalP 4.1 program predicted that the protein sequence presents a secretion signal peptide at the N-terminus between amino acids 1 to 17 (Appendix A). According to the results of the ProtParam tool, BASIDIN has a theoretical molecular weight of 17.9 kDa with the signal peptide, and 15.9 kDa after the removal of the signal peptide, which is carried out to enable expression of the protein in *Escherichia coli* (Appendix A). With the addition of six histidines (~1.7 kDa), the molecular weight of the protein is approximately 18.2 kDa.

The recombinant protein (rBASIDIN) revealed a band of approximately 20 kDa (Figure 1) and was expressed in the soluble and insoluble fractions of the *E. coli* extract. The molecular weight did not correspond to the expected value for the 6xHis-Tag fused recombinant protein, which can be explained by the disorder that the protein presents in its structure. rBASIDIN was recovered from the *E. coli* extract by affinity chromatography (Figure 1).

### 2.2. Bioassay with rBASIDIN on Tomato Leaves

rBASIDIN was inoculated on the leaves at a concentration of 4 µmol L^−1^ and caused curling starting at the leaf edges at 6 h after inoculation (hai) (Figure 2). Inoculation of leaves with the protein rMpNEPII at a concentration of 20 µmol L^−1^ caused necrosis restricted to the site of inoculation at 24 hai. The leaves inoculated with Tris-HCl buffer did not show any symptoms (Figure 2).

Tomato leaves inoculated by spraying with rBASIDIN at the concentration of 11 µmol L^−1^ showed symptoms of leaf shriveling at 2 hai, wilting and onset of necrosis at 6 hai, and total leaf drying at 24 hai (Figure 3).

### 2.3. H_2_O_2_ Production and Electrolyte Leakage of Tomato Leaves Sprayed with rBASIDIN

Discs from leaves inoculated by spraying with rBASIDIN at a concentration of 10 µg µL^−1^ and soaked in 3.3’ diaminobenzidine (DAB) showed more intense dark brown coloration due to H_2_O_2_ production compared to leaves that received no stress or were sprayed with Tris-HCl buffer (Figure 4A).

Leaves sprayed with a concentration of 4 μmol L^−1^ showed electrolyte extravasation of 12.9% (not significant), while the average of plants treated with buffer and no treatment was 10.6% (Figure 4B). Plants treated with a concentration of 10 μmol L^−1^ showed electrolyte extravasation values of 17.6%, significantly higher than that of untreated and buffer-treated plants (Figure 4B).

### 2.4. The Influence of rBASIDIN on Gas Exchange

During the three days of the experiment, the photosynthetic rate (Pn) values of tomato plants treated with rBASIDIN at 10 μmol L^−1^ decreased in comparison with the values of plants treated with the buffer (10 mmol L^−1^ Tris-HCl, pH 8.0), and this reduction was observed as early as 30 min after the protein was applied on the plants (Figure 5A and Appendix A).

Regarding stomatal conductance (gs), all plants showed similar behavior in both treatments, with a decrease after 90 min on all three days (Figure 5B and Appendix A). However, a significant decrease in gs occurred on the first day 20 min after rBASIDIN application, when the conductance value reached 0.08 mol m^−2^ s^−1^ (Figure 5B).

### 2.5. Circular Dichroism (CD) Analysis

The rBASIDIN protein showed a characteristic secondary structure predominantly of β-sheets, with a positive peak in the spectrum at 197 and a negative one at 217 ηm. The protein showed similar spectra at temperatures of 25 and 95 °C (Figure 6A).

Based on the results obtained by CD, leaves were treated with rBASIDIN at 11 μmol L^−1^, heated for 10 min at 80 °C, and then cooled to room temperature. Preheated rBASIDIN caused symptoms of leaf edge curling similar to that caused by the protein without prior heating (Figure 6B). Leaves treated with 10 mmol L^−1^ of preheated Tris-HCl buffer, like the protein-treated leaves, showed no symptoms of shriveling (Figure 6B).

### 2.6. Sensitivity Test of T. cacao to Recombinant Protein rBASIDIN

Cocoa leaves submitted to the application of rBASIDIN at 10 µmol L^−1^ showed tissue damage (Figure 7). Symptoms started five days after spraying, with chlorosis at the ends and margins of the leaf, becoming more evident after 10 days. At 20 days, the symptoms spread over the entire leaf, with the drying of the tissue in parts of the edge and in the region where the leaf was cut. Changes were not observed in control leaves, which remained green throughout the experiment (Figure 7).

### 2.7. Evolutionary Origin and Expression of the Effector Candidate BASIDIN

Sequence similarity-based analysis revealed that BASIDIN has orthology with several proteins annotated as hypothetical in other basidiomycete fungi. Despite this, searches for conserved sites indicated the absence of known conserved domains.

Phylogenetic analysis indicated that BASIDIN probably originated from basidiomycete fungi, with saprotrophic, hemibiotrophic, and parasitic lifestyles, and is most closely related to the genus *Armillaria*, showing strong node support in the branches, and to its sister taxon *M. roreri*. In addition, outer clusters were composed mostly of proteins that showed a higher degree of structural disorder (Figure 8).

Transcriptional analysis in the RNA-Seq libraries of the fungi with sequenced genomes confirmed that both the protein gene and its orthologs are transcribed. However, different transcriptional profiles were observed between BASIDIN and orthologous proteins, which can be visualized in the heatmap graphical representation, where expression values are represented by colors that vary as a function of expression intensity (Figure 8). In addition, we also observed differences in protein sizes and number of orthologs in the different fungal species evaluated (Figure 8).

To better understand the phylogenetic relationship of BASIDIN with homologous proteins grouped in the same clade, we performed alignment of the primary sequences of the proteins, which showed lengths ranging from 137 to 163 amino acids, but had a high conservation of residues among them, and no cysteine residues. In addition, they had a conserved loop region, varying only in size (Figure 9A).

The proteins were compared according to their biochemical characteristics, using some bioinformatics programs. In general, they were acidic proteins, with molecular mass ranging from 14 to 17 kDa, with a peptide signal. Except for PBK74827.1 from *Armilaria solidipes*, all showed a prediction of extracellular location (Figure 9B).

## 3. Discussion

### 3.1. Biochemical Characterization and Selection of BASIDIN

The criteria used to predict effectors were small proteins (≤250 aa) rich in cysteine, not having homology with proteins already characterized in the databases, not having conserved domains, and presenting a signal peptide for secretion. To predict the *M. perniciosa* effectorome, Barbosa et al. (2018) [7] used the effector prediction pipeline based on [16], which considers at least one of these effector-oriented criteria. In this context, we combined bioinformatics analyses to select the possible effector candidate to be functionally characterized in this work, taking into account the one that met most of the aforementioned criteria, and that presented a >50% probability of being an effector by the EffectorP-3.0 software. BASIDIN fulfills most of the criteria described by Toro and Brachmann (2016). It has 162 amino acids, does not have homology with proteins already characterized in the databases, does not have a conserved domain, and has a signal peptide for secretion (Appendix A). BASIDIN only lacks cysteine residues. It was predicted with an 84% probability of being an effector protein, so it was the effector candidate selected here.

The recombinant protein (rBASIDIN) revealed a band of approximately 20 kDa, but the molecular weight (Mw) did not correspond to the expected value for a 6xHis-Tag fused recombinant protein (Figure 1). This may have been associated with the protein’s structural disorder (Figure 8). Disordered proteins (IUPs) are known to have larger Mw than the actual one deduced from sequencing [17]. The reason for this is that IUPs are relatively richer in acidic residues that bind less to SDS and therefore move more slowly through the gel than globular proteins [18]. The Mw of rBASIDIN observed on SDS-PAGE was approximately 1.09 times greater than the Mw deduced from rBASIDIN (18.2 kDa). Therefore, the expression of rBASIDIN, linked to six histidines, was confirmed through gel-free mass spectrometry (LC-MS/MS).

### 3.2. BASIDIN Induces Wilting, Curling, and Necrosis Symptoms in Tomato Leaves

The action mechanisms of most effectors encoded by the genomes of phytopathogenic fungi have not been described in the literature, as is the case of the protein encoded by ORF Mp4145-3305, which we call BASIDIN, predicted in the genomic bank [7] of a subpopulation of the fungus *M. perniciosa* isolated from susceptible cocoa genotypes. *M. perniciosa* is hemibiotrophic, causing major losses in cocoa bean production in the Americas, but it has no specific effectors characterized, only a known necrosis and ethylene-inducing protein [19,20]. It was first identified in the culture filtrate of the *Fusarium oxysporum* isolate [21]. Necrosis-causing proteins facilitate the colonization of *M. perniciosa* in host tissues by inducing cell death upon shifting to the necrotrophic phase of the fungus [19].

rBASIDIN caused varied symptoms in tomato leaves, even when inoculated at low concentrations, compared to MpNEP2. It was able to cause leaf curling, suggesting it is a potent effector (Figure 2). When the concentration of rBASIDIN increased, more severe symptoms occurred in leaf tissues, such as curling, wilting, and necrosis, leading to total leaf drying (Figure 3).

The same symptoms characteristic of plant tissues undergoing the first signs of cell death have been reported when inoculating effectors in susceptible plants [19,22,23]. Moreover, in the transition from the biotrophic to the saprophytic phase in cocoa witche’s broom, leaf curling and wilting occur, and the overgrown (“broom-like”) branch dries rapidly, similar to the symptoms observed in rBASIDIN-treated tomato leaves.

Thus, cell death at the site of infection, typical of the hypersensitivity response [24,25], is a response triggered by the plant’s defense system through the recognition of a pathogen a virulence protein by a host resistance protein. On the other hand, BASIDIN can act as an effector for the necrotrophic phase of the disease, triggering generalized cell death, affecting the entire branch. This can be proven by the transient transformation of the probable effector BASIDIN in tomato plants, or the search for the targets of this effector in the plant by analyzing the protein–protein interaction by means of a double-hybrid assay [26] or by target capture assays by resin-immobilized rBASIDIN, followed by identification of the captured host proteins by mass spectrometry.

### 3.3. BASIDIN Causes Damage to Leaf Membranes and Induces Oxidative Stress in Tomato Plants

Hydrogen peroxide production is recognized as the first response to biotic stress in plants, often causing irreversible damage to cells [27]. Thus, to validate the function of the protein for this particular response, we used Micro-Tom model plants, due to their short life cycle, small size, and ease of in vitro analysis, as well as having compatible interaction with *M. perniciosa* and developing symptoms similar to those of witche’s broom in cacao [3].

The Micro-Tom plants sprayed with rBASIDIN showed a slight accumulation of H_2_O_2_ in comparison with the controls (Figure 4A). However, high levels of ROS can act as signaling molecules that activate multiple defense responses in the plant, resulting in programmed cell death [28]. In turn, at low concentrations, ROS induces an adaptive response [29].

However, because of the hemibiotrophic life cycle of *M. perniciosa*, its secretion of an effector protein that will intensify ROS (reactive oxygen species) formation after PAMP recognition is a very relevant adaptive strategy for the pathogen, since this results in a larger necrotic lesion, benefiting its necrotrophic stage [9]. Interestingly, an in vivo study demonstrated that the combination of low concentrations of H_2_O_2_ and glycerol induced the dicariotization of hyphae of *M. perniciosa*, resulting in a change from the biotrophic to necrotrophic phase and that the fungus tolerates high concentrations of hydrogen peroxide in the culture medium, through the action of peroxidases and catalases [30].

The increase in peroxide accumulation caused by BASIDIN in tomato leaves was not enough to claim that it regulates ROS production. On the other hand, some fungi secrete effector proteins to suppress the production of ROS as a strategy to overcome the host’s innate immunity [31]. For example, *Phytophthora* species express the effector PpCRN7 to decrease hydrogen peroxide accumulation by negatively regulating the host plant’s defense-associated genes [9].

Damage to the membrane and cell wall may also be one of the first signs of stress in the plant. One way to verify this type of stress is to analyze electrolyte leakage, taking into account that the less stressed the plant is, the less the membranes are ruptured and the lower the level of electrolyte leakage [27]. Tomato leaves subjected to rBASIDIN treatment showed damage to membranes, demonstrated by a significant increase in extravasation levels compared to controls, and this increase was proportional to the concentration of the protein (Figure 4B). This suggests that the protein may be involved with the extravasation of cellular electrolyte content. This type of stress can result in cell death [27,32]. Symptoms of cell death caused by the SCRE2 effector in *Nicotiana benthamiana* plants were highly correlated with the percentage of electrolyte leakage [33].

### 3.4. BASIDIN Causes Photosynthetic Inhibition of the Plants

rBASIDIN increased the photosynthetic rate of tomato plants 30 min after application and reduced it in the following intervals, with its lowest rate at 120 min, when wilting occurs, as shown by our results (Figure 3 and Figure 5). These findings suggest that the reduction in the photosynthetic rate occurred due to the impairment of biochemical (non-stomatal) factors. The reduction of photosynthesis by biochemical factors can occur due to impairment caused by the pathogen to foliar biochemical reactions [24], and biochemical alterations, with the decline of the photosynthetic rate resulting in leaf senescence and tissue necrosis [6].

However, it was possible to observe a significant increase in the photosynthetic rate in the tomato plants in the control condition at a time of 30 min, after which the photosynthesis rate tended to stabilize until 60 min, reaching almost its baseline photosynthetic rate, as shown in Figure 5. This type of response is common in plants under homeostatic conditions (control), which in natural conditions rhythmizes and regulates the circadian cycle. Several biological processes in plants are controlled by the endogenous circadian clock, such as gas exchange and chlorophyll fluorescence parameters, as seen in isogenic strains carrying wild-type allele combinations in tomato [34,35], in a study of *Arabidopsis thaliana*, tested the role of the circadian clock in the function of phothoystem II and photoprotection. Their results demonstrated significant variation for all photoprotective components, suggesting that the circadian cycle undergoes adaptations and derives in part from its regulation of the light reactions of the photosynthesis and photoprotective mechanisms.

Many pathogens secrete effectors targeting the host plant’s chloroplasts, causing damage to the thylakoid membrane and suppressing the production of defense signals [36,37]. The effector Pst_12806 of the fungus *Puccinia striiformis* f. sp. *tritici*, when translocated into the host cell’s chloroplast, interacts with the functional domain of the TaISP protein and decreases the ability of this domain to transfer electrons during the photosynthesis process of wheat plants [37]. *A. thaliana* plants inoculated with a *Pseudomonas syringae* strain possessing the avirulence effector gene PstI-AvrRpt2 activated two kinases (MPK_3_ and MPK_6_) in the plant that act by inhibiting photosynthesis to increase resistance to the pathogen. Therefore, it has been suggested that inhibition of photosynthesis triggered by effectors contributes positively to the activation of ETI, resulting in ROS accumulation in the chloroplasts and programmed cell death [38].

However, our results obtained during gas exchange measurements suggested that BASIDIN may be involved in impairing photosynthesis. Thus, the next step to validate this hypothesis was to analyze which mechanism the protein uses to promote a negative effect on the photosynthetic rate of the plants.

### 3.5. BASIDIN Is Thermostable and Rich in β-Sheets

The effectors are usually divergent in sequence due to high selection pressure to avoid recognition by the host, so it is often challenging to perform their functional characterization [39]. However, information obtained through structural analyses of effector proteins is relevant for characterization by offering clues about their biochemical identity and how they interact with the host cell [14,39].

The CD spectra of the protein revealed a predominantly β-sheet structure, with a positive absorbance peak at 197 ηm and a negative one at 217 ηm (Figure 6A), compatible with the standard spectra expected for this structure by the oligo-polypeptide model [40]. This result is in agreement with the data from in silico analysis, which predicted a higher percentage of β-sheets in the protein structure (Appendix A). Secondary structure analysis of effector proteins has revealed the abundant presence of alpha-helices in several of these molecules [41,42], such as in the RxLR family of effectors, in which the abundance of alpha-helices supports their functional diversity by allowing for deletion and insertion of amino acids in their structures without compromising their functions [43]. In silico analysis predicted a 24.07% alpha-helix level in the secondary structure of the candidate effector protein (Appendix A).

Thermophilic proteins generally have a higher alpha-helix and beta-sheet content [44,45], so the beta-sheet conformation of BASIDIN may contribute to its thermal stability since the spectrum at 25 °C was equivalent to that at 95 °C (Figure 5B). Furthermore, the protein maintained its activity even after being subjected to 80 °C for 10 min (Figure 6B). This suggests that the β-sheet structure of rBASIDIN is organized into beta aggregates strongly stabilized by intermolecular interactions [46]. A similar profile was reported for some NEP proteins of *M. perniciosa*, whose activities remained stable even after treatment at 100 °C [19].

### 3.6. rBASIDINA Induced Symptoms in Cacao

*T. cacao*, the main host of *M. perniciosa*, showed symptoms of chlorosis, with the death of leaf tissue 20 days after application of rBASIDIN on the leaf (Figure 7), as seen in the Micro-Tom model plant (Figure 7), which is also a host of *M. perniciosa*. The results suggest the existence of a conserved mechanism for recognizing this effector, triggering deleterious symptoms in the plant, corroborating the existence of effector proteins in the evolution of witches’ broom disease caused by the phytopathogen *M. perniciosa* [7,11,47,48]. In addition, effects similar to those caused by rBASIDIN in non-inoculated *T. cacao* leaves also occurred in leaves inoculated with MpNEPI [49]. In this study, we found the presence of chlorosis symptoms, followed by a brown lesion (necrosis) from the edge to the center of the leaves when sprayed with MpNEP1, a protein of *M. perniciosa*, inducer of ethylene and necrosis [19].

Application of rBASIDIN at the same concentrations to cocoa and tomato leaves caused milder symptoms in cocoa. This can partly be explained by the morphoanatomical characteristics of cocoa leaves, which have a leathery texture and are hypostomatic, with the occurrence of a few stomata near the main vein on the abaxial surface [50], unlike tomato leaves. This may interfere with the ability of the leaf to absorb rBASIDIN, leading to the generation of milder symptoms in comparison with those that occur in Micro-Tom, which has membranous leaves and an average number of stomata per mm^2^ of 538 in the adaxial epidermis and 881 in the abaxial epidermis [51].

The hypothesis of greater efficiency of the effector in Micro-Tom is due to differences in the biochemical mechanisms of action, from the interaction with possible receptors in the plants in question, leading to a faster response (2 h) when exposed to the effector. Further studies may be carried out for this investigation. Taken together, these findings contribute to understanding the mechanisms that determine the compatible reaction between rBASIDIN and non-host plants or even the different levels of resistance to witches’ broom disease.

### 3.7. BASIDIN Is a New Basidiomycete-Specific Effector Protein

Phylogenetic analysis indicated that BASIDIN is related to proteins from basidiomycete fungi, and the vast majority of these fungi are phytopathogenic [52,53,54], except *Bondarzewia mesentericum*, *Hericum alpestre,* and *Pheurotus ostreatus*, which are primary decomposers, causing white rot in wood [55]. However, they are prized for their nutritional and medicinal properties [56,57]. In addition, the cultivation of these fungi makes it possible to economically recycle agricultural waste, due to their production of extracellular enzymes such as laccases and proteases [57,58,59]. This group also contains the brown-rot fungi *Gloeophyllum trabeum*, *Neolentinus lepideus,* and *Heliocybe sulcata*, which mainly attack softwood trees like conifers and pines [60]. The decomposer activity of brown-rot fungi causes great harm in the northern hemisphere, since most of the wood used in construction is from coniferous species [61]. The orthologous proteins are deposited in the database as hypothetical proteins, and have no known functional domains that could help infer a likely function for this group of proteins. However, the literature reports that these basidiomycetes express many genes considered important to facilitate infection or obtain nutrients [7,62,63,64].

The genus *Armillaria*, for example, which is considered a pathogen of several woody, ornamental, and fruit plants of economic importance [52,65], colonizes the host by expressing effector proteins that promote changes in the plant, causing *Armillaria* symptoms such as root rot, leading to death or poor crown development. *Armillaria ostayae* is considered the most virulent species [52,63].

Another curious fact, reported by some researchers, is the sharing and similarity of several virulence genes between the fungi *M. roreri* and *M. perniciosa*, among them the effector NEP1 and effectors rich in cysteine residues called hydrophobins [7,53,54]. Likewise, BASIDIN showed higher similarity and identity with a protein from the pathogen *M. roreri*, sharing the same branch in the phylogenetic tree (Figure 8).

The sequence alignment demonstrated some conserved positions in the protein sequences, suggesting that these sites are important for maintaining function and conformation since the functional sites of the proteins tended to be more conserved. In addition, transcriptome analysis showed that BASIDIN and its orthologs are expressed, corroborating that these proteins are involved in some biological processes important for the establishment and progression of the disease in their hosts.

In this context, the degree of intrinsic disorder in protein sequences was investigated since this is an important feature in effectors because of its relation with function and protein–protein interaction [66,67,68].

The clade containing the proteins PBK97726.1, PBK74820.1, PBK74827.1, SJL07483.1, BASIDIN, and ESK95757.1 of *Armillaria gallica*, *Amillaria solidipes*, *A. ostayae,* and *M. roreri* was formed by partially disordered proteins, possessing between 10 and 20% disordered residues. Most of these residues were found in a conserved loop region (Figure 8 and Figure 9A). The presence of a disordered loop was also observed in the ATR13 effector sequence of the oomycete *Hyaloperonospora arabidopsidis*, in which this region was highly flexible and involved in the nuclear localization of the effector in the host cell [66]. In the most basal group and the outer group, we observed the proteins with the highest degree of disorder, with 26 to 38% disordered residues (Figure 8).

Comparative analyses of the proteins that were phylogenetically closest to BASIDIN showed they have structural characteristics of effector proteins, are small (ranging from 137 to 163 amino acid residues), have no apparent functional domain, have signal peptide in the N-terminal region, and lack cysteine residues (Figure 9B), which is an exception to the rules for effector proteins. However, the same pattern was observed in the sequence of the effector *AvrM* from the fungus *Melampsor alini* [69].

Considering that an effector protein that is shared among the same species or that has homologs in different species is considered a central effector, these analyses allow us to suggest the existence of a group of proteins with potential molecular and evolutionary characteristics, which can be exploited to monitor the pathogenicity of various pathogens of the basidiomycete group.

In conclusion, the findings of this work suggest that the protein BASIDIN is a new effector of the fungus *M. perniciosa*, with good biotechnological potential due to its biochemical characteristics, such as stability at high temperatures and its biological effect on plant immunity. These characteristics can support future studies to investigate the contribution of BASIDIN to the activation of plant-induced immunity, as well as its differential expression in the pathosystem (*M. perniciosa* × *T. cacao*). Furthermore, other homologous proteins may play a similar role, which can contribute to plant genetic improvement.

## 4. Materials and Methods

### 4.1. Selection of BASIDIN

The Mp4145-3305 effector candidate protein (BASIDN) was identified in subpopulations of the fungus *M. perniciosa* (Biotype C) in the work of Barbosa et al. The complete sequence of BASIDIN amino acid residues was subjected to bioinformatics analysis to confirm the prediction as an effector.

For this, the presence of signal peptide was evaluated using the SignalP 4.1 program (http://www.cbs.dtu.dk/services/SingnalP/ (accessed on 18 September 2018)) and similarity analysis using the BLASTp program. The verification of possible post-translational modification sites was made using the following programs: NetAcet1.0 server (http://www.cbs.dtu.dk/services/NetAcet-1.0/ (accessed on 18 September 2018)) to identify possible acetylation sites, and the NetNglyc 1.0 Server (http://www.cbs.dtu.dk/services/NetNGlyc/ (accessed on 18 September 2018)), to identify possible N-glycosylation sites. Furthermore, the BASIDIN sequence was submitted to the EffectorP program (https://effectorp.csiro.au/ (accessed on 18 September 2018)), which predicts effectors.

### 4.2. Expression of BASIDIN in Escherichia coli

The *E. coli* strain Rosetta (DE3) was transformed with the pUESC 251 BASIDIN vector, a pET-28a construct carrying the synthetic ORF encoding BASIDIN at the Nde I/Xho I a site (Biomatik, Cambridge, ON, Canada).

BASIDIN’s synthetic ORF was constructed without the signal peptide, and for expression in *E. coli* it was optimized for rare codons, without compromising the protein sequence (Appendix A). In addition, a histidine tail was added to facilitate the purification process. Protein induction was performed in LB (Luria-Bertani) medium at 37 °C with 0.4 mmol L^−1^ of isopropyl beta-D-thiogalactoside (IPTG) for 4 h at 180 rpm. Recombinant BASIDIN (rBASIDIN) was purified by affinity chromatography using a His Trap FF crude nickel column (GE Healthcare, Chicago, IL, USA), dialyzed, and quantified by the Bradford method (1976). The identity of recombinant protein was confirmed by mass spectrometry.

### 4.3. Bioassay with Tomato Leaves (Micro-Tom)

A volume of 20 μL of rBASIDIN at 4 μmol L^−1^ in 5 mmol L^−1^ of Tris-HCl pH 8.0 was injected into the abaxial part of detached leaves of the tomato plants (*S. lycopersicum* cv Micro-Tom) at 8 weeks after planting, near the central vein.

Leaves from control plants were injected with 5 mmol L^−1^ of Tris-HCl buffer pH 8.0. The recombinant necrosis-inducing protein and ethylene of *M. perniciosa* (rMpNEP2), at the same concentration as BASIDIN and a concentration of 20 μmol L^−1^, were injected into Micro-Tom leaves, as described by Garcia et al. (2007) [19].

After inoculation, the Micro-Tom leaves were separately placed in Petri dishes lined with moistened filter paper to prevent dehydration. Images of the leaves were captured at 0, 6, 12, and 24 h post-inoculation.

As a way to further evaluate leaf stress by the action of the protein, one-month-old *S. lycopersicum* plants were sprayed with the recombinant protein BASIDIN at a concentration of 11 μmol L^−1^ in Tris-HCl 10 mmol L^−1^. The concentration was defined based on preliminary tests that ranged from 1 µmol L^−1^ to 11 µmol L^−1^ (Appendix A).

Symptoms were also recorded by photographs. The control treatment consisted of plants sprayed with 10 mmol L^−1^ of Tris-HCl buffer.

### 4.4. Application of rBASIDIN on Theobroma cacao Leaves

The experiment was carried out under greenhouse conditions, using young *T. cacao* plants of the Catongo (susceptible) genotype, 27 days after seed planting. The third part of the leaf from the apex was removed and the effector protein rBASIDIN (10 µmol L^−1^ in 10 mmol L^−1^ sodium phosphate buffer) was sprayed onto the abaxial face of the remaining 2/3 of the leaf. As a negative control, the last dialysis buffer sodium phosphate 10 mmol L^−1^ was used. Leaf protein activity was monitored by recording photographic images for 20 days.

### 4.5. H_2_O_2_Accumulation Test

To evaluate the involvement of BASIDIN in the generation of ROS in tomato leaves, the accumulation of hydrogen peroxide was analyzed. Two leaves of *S. lycopersicum* were sprayed with rBASIDIN, and as a control, two leaves were sprayed with 10 mmol L^−1^ of Tris-HCl buffer pH 8. At 6 h after application, leaf discs were collected and immersed in 1 mg mL^−1^ of 3-3′ diaminobenzidine (DAB) and HCl, pH 3.8. Next, the leaf discs were subjected to vacuum for approximately 30 min, after which they were incubated in the dark for 24 h. Afterward, the discs were boiled in 96% ethanol for 20 min and then washed with 50% ethanol until all chlorophyll was removed. The images were recorded with a Leica EZ4D magnifier and photographed using the program LAS EZ (Leica Application Suite, Version 1.4.0). The experiment also had vacuum infiltration of leaf discs with autoclaved water as a control.

### 4.6. Electrolyte Extravasation

To evaluate whether rBASIDIN affects the stability of leaf membranes, electrolyte leakage analysis was performed [70]. Three plants were used for each treatment and one leaf from each plant was sprayed with 500 µL of rBASIDIN solution at concentrations of 4 µmol L^−1^, 10 µmol L^−1^, or Tris-HCl 10 mmol L^−1^. Plants with no treatment applied were also considered controls. Four hours after the application of the treatments, 10 leaf discs, each 1 cm in diameter, were collected and washed in distilled water to remove the contents of the cells that ruptured during removal and other electrolytes adhered to the leaves. After washing, the leaf discs were dried on absorbent paper and then placed in falcon tubes containing 10 mL of ultrapure water at 25 °C for 24 h under constant stirring. Subsequently, the initial conductivity of the medium where the disks were located (C1) was measured with a Delta OHM bench top conductivity meter (HD.2106.1). After this procedure, the tubes with the leaf discs were placed in a water bath at 90 °C for 2 h. After reaching room temperature, the maximum conductivity was measured (C2) and the electrolyte leakage was calculated using the formula: [(C1 − B1)/(C2 − B2)] × 100. Tubes similar to the samples containing only ultrapure water were used as controls (B1) before boiling and (B2) after boiling. The data were analyzed, and the means were compared by the Tukey test (*p* < 0.05).

### 4.7. Gas Exchange

For gas exchange analysis, a portable photosynthesis analyzer was used Li-Cor Li-6400 XT (LiCor®, Lincoln, Nebraska, USA). rBASIDIN at the concentration of 10 μmol L^−1^ was applied to the third pair of leaflets of the second leaf from the stem apex, and five tomato plants were used for the treatment. The control group also consisted of five plants, which were treated with 10 mmol L^−1^ of Tris-HCl buffer. After 30 min, spot measurements of gas exchange were made on the leaves, using a reference CO_2_ injector at 400 μmol mol^−1^ under irradiance of 1000 μmol photons m^−2^s^−1^ and fixed block temperature of 27 °C. Data collection always started at 8:00 a.m., and four readings were taken throughout the day, every 30 min, in the period between 8:00 and 9:30 a.m. The rBASIDIN concentration was based on preliminary test results for spraying bioassay analysis (Appendix A).

### 4.8. Circular Dichroism (CD) Analysis

The circular dichroism spectrum of rBASIDIN was collected from 190 to 240 ηm using a J-815 spectropolarimeter (Jasco), with a thermostatically controlled stand with a PTC-423S/15 Peltier device. For this, we used 300μLof the protein at a molar concentration of 0.05 μmol L^−1^; in Tris-HCl buffer 5 mmol L^−1^, pH 7.4. The scans were made with the Spectra Manager program (Jasco), at a speed of 50 ηm min^−1^ and a collection interval of 1.0 ηm. Readings were taken at temperatures of 25 and 95 °C, using a polarized quartz cuvette with a 1 mm path length. The spectra obtained were the results of six consecutive scans, and the data from the spectra were analyzed with the K2D3 program to estimate the percentage of the secondary structure of the recombinant protein rBASIDIN.

### 4.9. Phylogenetic Analyses

For phylogenetic analysis, proteins related to the BASIDIN effector candidate were selected from BLASTp against the NCBI non-redundant (NR) database and submitted to subsequent correlation analysis [71] (Appendix A). The level of conservation among the selected proteins was verified from multiple alignments using the ClustalW from MEGA 7 software [72], and Clustal Omega online [73]. All selected proteins were subjected to domain and conserved site analyses from the HMMER-EMBL-EBI [74] and Protein Residue Conservation Prediction [75] platforms.

The primary sequences of the proteins under study were submitted for the prediction of possible disorders in their structure using the predictors PONDR VL-XT [76] and IUPred [77]. The output of the IUPred results was generated using the “long disorder (default)” option.

Evolutionary analysis was performed using MEGA 7 [72] from alignment with 21 candidate effector protein sequences. The tree was constructed using the neighbor-joining method [78], with 1000 bootstrap replications [79]. Evolutionary distances were calculated by the JTT matrix method [80]. All positions containing gaps and missing data were excluded from the analysis.

### 4.10. Analysis of the Transcripts

The reference sequences of the transcripts referring to the genomes of the investigated fungi (*A. gallica*, *A. solidipes*, *A. ostoyae*, *M. roreri*, *B. mesenterica*, *H. alpestre*, *G. trabeum*, *N. lepideus*, *H. sulcata*, *G. luxurians*, *C. laeve*, and *P. ostreatus*) were obtained from the GenBank (https://ftp.ncbi.nlm.nih.gov/genomes/genbank/fungi/ (accessed on 4 December 2020)), in fasta format (Appendix A). Transcripts from each fungus were evaluated according to their transcriptional profile on the RNA Galaxy workbench 2.0 platform (https://rna.usegalaxy.eu (accessed on 4 December 2020)) using libraries of long RNAs from each fungus obtained from the public SRA RNA-Seq database at NCBI (https://www.ncbi.nlm.nih.gov/sra/ (accessed on 4 December 2020)) (Appendix A). Quantification analysis was performed with the Salmon tool (version 1.9.0) [81] according to parameters preset by the software. The results obtained made it possible to determine the relative quantification of each transcript normalized by transcripts per million (TPM). For better visualization of the transcript expression profile, the transcript quantification data were plotted in heatmap form.

## Figures and Tables

**Figure 1 ijms-24-11714-f001:**
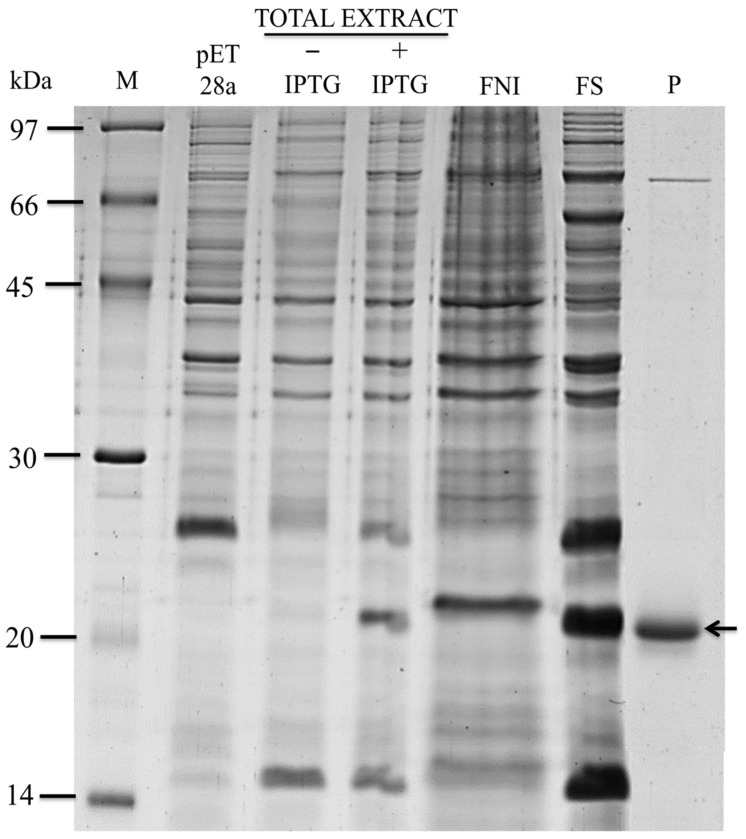
Expression, solubility, and purification testing of BASIDIN. Expression of recombinant BASIDIN was performed in *E. coli* Rosetta (DE3) at 37 °C. To follow in detail the process of recombinant protein expression, aliquots of the bacterial culture were collected before induction and 4 h after the addition of IPTG. Samples were analyzed in 12.5% SDS-PAGE. M corresponds to a standard molecular mass in kDa; pET28a denotes pET-28a without the insert; -IPTG corresponds to bacterial lysate before induction of rBASIDIN protein expression; +IPTG denotes bacterial lysate 4 h after rBASIDIN induction; SF stands for the soluble fraction; and IF the insoluble fraction. The arrow indicates protein bands of approximately 20 kDa corresponding to rBASIDIN with the addition of the histidine tail derived from pET-28a, purified from the soluble fraction.

**Figure 2 ijms-24-11714-f002:**
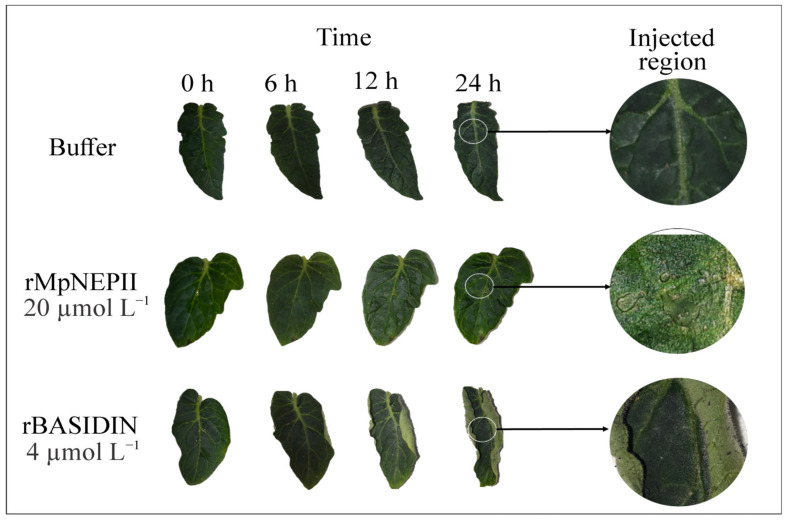
Inoculation of rBASIDIN and rMpNEPII on *Solanum lycopersicum* (tomato) leaves. Images were recorded at 0, 6, 12, and 24 h after inoculation (hai) with 4 μmol L^−1^ of the candidate effector protein rBASIDIN, with Tris-HCl buffer (negative control) and with MpNEPII at 20 μmol L ^−1^ (positive control). The expanded image focuses on the region of inoculation with MpNEPII on a tomato leaf, highlighting the site where necrosis symptoms occurred, and the 4 μmol L^−1^ inoculation site with no necrosis, similar to that observed with Tris-HCl buffer.

**Figure 3 ijms-24-11714-f003:**
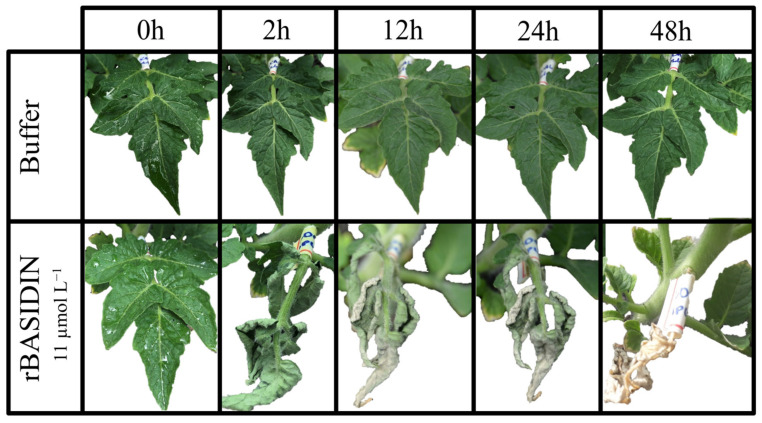
Images of tomato plants with leaflets sprayed with rBASIDIN. The control plant sprayed with 10 mmol L^−1^ Tris-HCl buffer, pH 8.0, shows no symptoms. The plant sprayed with rBASIDIN shows leaf shriveling at 2 h, onset of necrosis and wilting at 12 h after treatment, and complete drying of the leaves after 24 h.

**Figure 4 ijms-24-11714-f004:**
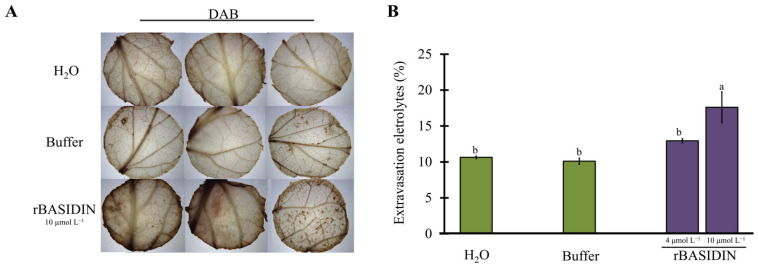
Hydrogen peroxide production and electrolyte leakage test. (**A**): Hydrogen peroxide production in leaf tissue of *S. lycopersicum* sprayed with rBASIDIN. Leaf discs were taken from plants subjected to the treatments and H_2_O_2_ production was visualized by staining with DAB. Plants were vacuum inoculated with 10 mmol L^−1^ of 3,3′ diaminobenzidine for 30 min. Water: plants that did not undergo any stress; Buffer: plants that were stressed for 6 h with Tris-HCl 10 mmol L^−1^ from the last dialysis; plants that were stressed for 6 h with rBASIDIN at a concentration of 10 μmol L^−1^. (**B**): Analysis of electrolyte leakage in *S. lycopersicum* plants sprayed with different concentrations of rBASIDIN. Columns in the graphs indicate an average of 3 repetitions (n = 3). Columns followed by the same letter do not differ by the Tukey test (*p* < 0.05).

**Figure 5 ijms-24-11714-f005:**
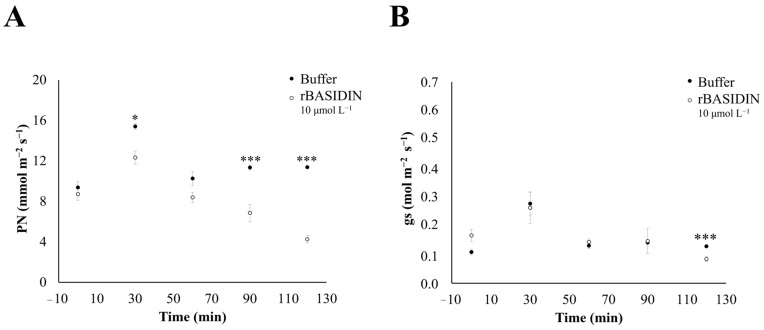
Variation of gas exchange of *Solanum lycopersicum* plants treated with rBASIDIN. Readings are taken from 8:00 to 9:30 a.m. (**A**): Photosynthetic rate. The filled circles (●) correspond to the photosynthetic rate of plants treated with 10 mmol L^−1^ of Tris-HCl buffer, pH 7.0. The empty circles (○) correspond to plants treated with the recombinant protein BASIDIN at a concentration of 10 μmol L^−1^. Error bars represent the standard deviation of the mean (n = 5). The treatments showed significant differences by ANOVA at 5%. (**B**): Stomatal conductance. The filled circles (●) correspond to the conductance rate of plants treated with 10 mmol L^−1^ of Tris-HCl buffer, pH 7.0. The empty circles (○) correspond to treatment with the recombinant protein BASIDIN at a concentration of 10 μmol L^−1^. Error bars represent the standard deviation of the mean (n = 5). The treatments showed significant differences by ANOVA at 5% only at 120 min. Probability level (P) indicate a significant difference in photosynthetic rate and stomatal conductance between plants treated with rBASIDIN versus control. (*) and (***) significant at *p* ≤ 0.05 and *p* ≤ 0.001, respectively.

**Figure 6 ijms-24-11714-f006:**
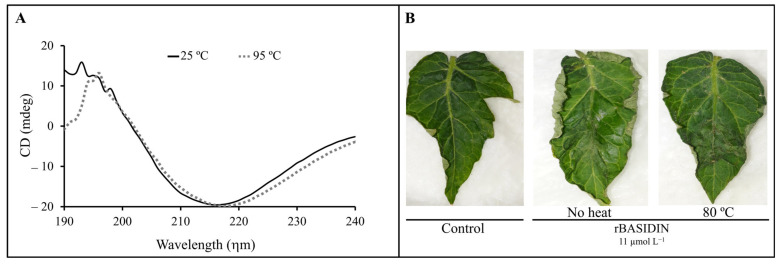
Circular dichroism spectrum of rBASIDIN. The scanning took place between the wavelengths of 190–240 ηm. The protein showed a positive peak at 195 ηm and a negative one at 217 ηm. The CD spectra were obtained at 25 °C and 95 °C. The spectra showed that the protein does not lose its conformation when treated at 95 °C. The black line shows the scan profile at 25 °C; the gray line shows the scan profile at 95 °C. The rBASIDIN protein concentration was 0.05 μmol L^−1^ (**A**). Visualization of rBASIDIN protein activity upon heating. The protein was inoculated in the leaf after remaining for 10 min in a water bath at 80 °C. The image represents the result of 24 h of inoculation, showing that the protein maintained its activity (**B**).

**Figure 7 ijms-24-11714-f007:**
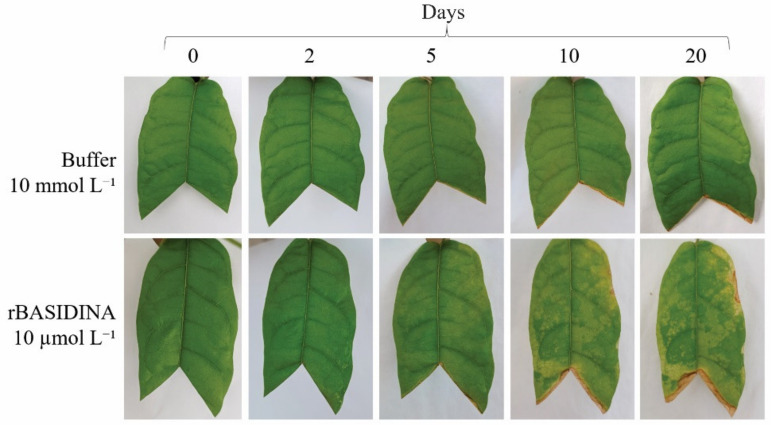
Activity test of rBASIDIN in *T. cacao* leaves. In the first row, *T. cacao* leaves sprayed with sodium phosphate buffer 10 mmol L^−1^, negative control; in the second row, *T. cacao* leaves sprayed with rBASIDIN 10 µmol L^−1^. Images recorded for 20 days (the duration of the experiment) after spraying the treatments.

**Figure 8 ijms-24-11714-f008:**
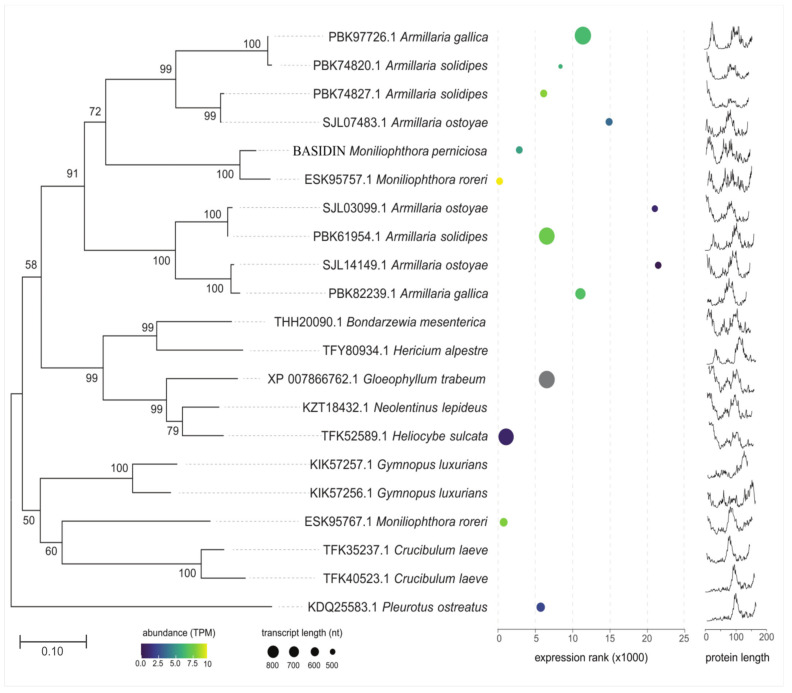
Evolutionary and molecular characterization of BASIDIN. Dendrogram obtained by neighbor-joining from the amino acid sequences of 21 proteins homologous to BASIDIN. The bootstrap values (1000 replicas) are indicated at the nodes of the tree. The bottom bar in the figure is related to genetic distance. In the middle panel, the circumferences represent the size of BASIDIN transcripts and homologous sequences in the different fungi. The color refers to the transcriptional quantity of each protein, ranging from dark blue to yellow, from lowest to highest expression, respectively. The X-axis represents the expression ranking of the transcript relative to all transcripts of that fungus. The graphs in the right panel indicate disorder analysis of the evaluated proteins.

**Figure 9 ijms-24-11714-f009:**
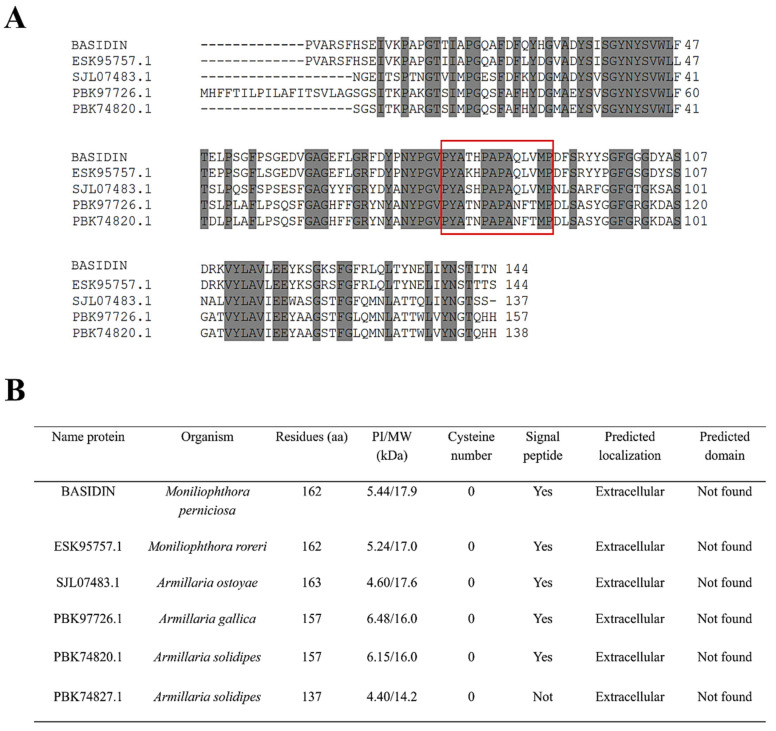
Alignment of BASIDIN with the proteins that were in the same cluster. In gray are the conserved regions in both proteins; the red box highlights the conserved loop region in all five proteins (**A**). Molecular comparison of the 5 proteins. The physicochemical parameters were predicted by the Protparam analytic program. For signal peptide prediction, the program SignalP 4.1 was used. All but one protein had a signal peptide between residues 1 to 23, suggesting that such proteins are secreted (**B**). Domain composition analysis by Pfam and InterProScan showed that all proteins have no known conserved domains in the databases. Localization prediction made using the DeepLoc-1.0 software demonstrated that all proteins have an extracellular location.

## Data Availability

The data presented in this study are available in the article and Appendix A (https://github.com/submissao10/Supplementary-MDPI_Farias-K.S..git, accessed on 16 July 2023).

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
