# Peer review of "BASIDIN as a New Protein Effector of the Phytopathogen Causing Witche’s Broom Disease in Cocoa"

_ijms, 2023, doi:10.3390/ijms241411714_

Round 1

Reviewer 1 Report (Previous Reviewer 3)

The revised manuscript by Keilane Silva Farias and colleagues has been improved considerably. Although it is generally convincing that this protein is involved in the described phenotypes, however, in my opinion and as I also stated in my previous reports, essential in order to support any conclusions is the proof that this protein is indeed secreted and also expressed during infection.

Reviewer 2 Report (Previous Reviewer 1)

Dear authors,

Happy with the outcome, despite difficulties, on the host plant. In confirms more pronounced activities on other plants which is a great contribution to the manuscript. With that, the manuscript can be published.

Minor point:  Barbosa et al, line 498 needs a year.

There are some "hick-ups" in English which might be checked briefly by a native speaker. 

This manuscript is a resubmission of an earlier submission. The following is a list of the peer review reports and author responses from that submission.

Round 1

Reviewer 1 Report

The manuscript by Farias et al., entitled “BASIDIN as a new protein effector of the phytopathogen causing witch’s broom disease in cocoa” describes the characterization of the protein BASIDIN for its potential function in witches’ broom disease. The manuscript in general is well written, some textual comments are included in the “other points raised” section below. There are a few points that are of more importance though.

First of all, the selection of this protein is only mentioned in the introduction but actually, there is no further information why it was selected in the result section. That raises questions and it would be valid to request additional reasoning in the result section to select this protein.

Then, the protein/gene has been retrieved from a fungus that is causing WBD of the cocoa tree. All the work is performed only on tomato, so what is actually the biological relevance. Has there been any test on cocoa? If so, include the information, if not, why? Which type of the fungus (S or C) is used as reference here (only discussed that they exist in line 264, not indicated which) and are there changes (AA) in the protein that might be relevant.
Another major point is the production of the recombinant protein, what was produced exactly, the full protein, a “cleaved” version, with or without signal peptide? In the figure 1 legend suddenly a Histidine tail is mentioned but no further information is provided in the manuscript. The protein of 162 residues indicated in the text (GenBank: EEB89748.1) , line 79) should be (with signal peptide) 17.81KD, with the addition of a His(6*), you get at 18.63 kD but in Figure 1, the band is according to legend 20 KD, but in fact is slightly bigger. If you remove the signal peptide (which should be extremely logic for a secreted protein) and add a His(6), the difference gets even bigger (should be only 16.82 kD). This should be addressed in full detail what has been done!!
There seems some mix-up in concentrations which should be carefully addressed, e.g. in the abstract concentrations are mentioned as “4 μmol uL-1 & 11 μmol uL-1”, in line 101 it suddenly is 4 μmol L-1 (and line 114: 11μmol L-1 so from µL-1 to L-1?). In the Materials & Methods section, Basidin is used at 3.75 μL L-1 in 5 mmol L-1 which is quite a shift from molars to liters description, and 11.7 μmolL-1 (Line 427, no space included)

With the supplemental Figure 1 (assume this should be the file “Supporting Information 29299313_cluster7.pdf), literally there was no clue what I was looking at.. Please provide an explaining legend if even this is indeed the correct figure

General remark, shouldn’t all names be written with Italics? Noticed that they are not throughout the whole manuscript.

Other points raised:

Line 21, if stating for one treatment (spray) a time interval, then also provide a time interval for the wilting effect in the abstract.

Line 25, should read: that it has

Line 27, “in different species” this is extremely unclear.. are we talking about the fungi or plants.

Line 27, expressed & important, the importance is not directly linked to levels of expression in my opinion..

Line 27, should read: that this protein

Line 50, should read: to unravel the mechanism

Line 104, why was the experiment not continued for a longer time and in addition, the pictures in Figure 2 are quite difficult to interpret (e.g. the zoom in’s).

Line 123, please add in the text that you stain on H2O2 (title only tells it is produced, no information what is actually done)
Line 126, please mention that you spray with the protein, you could spray with anything at a concentration of 4 umol L-1.

Line 135 and several other occasion, why is BASIDIN suddenly without capitals. Are we still talking about rBASIDIN?

Line 146-147, please provide here or in the discussion a reason for the initial increase after 30 min.

Line 208, double point at the end of sentence

Line 236, what is “it” referring to? To Nep or BASIDIN

Line 242 & general remark, we had the results, in the discussion one should not refer to figures..

Line 254, was can?

Line 280, Besides the Greek mythology that is known, there is no other mentioning in the manuscript on EROS?

Line 353, Start of paragraph with “this group”, is difficult to address what is meant..

Line 357, wondering if hypothetical proteins or protein should be used here... I guess that these proteins are deposited as individuals so as protein.

Line 401, “ suggest that BASIDIN as a novel effector” misses a word

Line 407, elaborate on “synthetic ORF”, meant as produced rather than cloned, or also optimized for E. coli? (if so, provide the optimized sequence)

Line 467, why is the degree sign underlined?

Line 479, new protein? rBasidina...

Line 481, what is selected meaning here? Was there a selection made among proteins that show homology or is this just proteins with homology were derived by blasting the NCBI databast using Blast.. which blast was used, tBLASTn or BlastP only, please define!!

In the Supporting information, the text is not always in English, please check and correct.

Author Response

We sincerely appreciate yours contribution in the review and suggestions regarding the manuscript "BASIDIN as a new protein effector of the phytopathogen causing
witch’s broom disease in cocoa
." (Manuscript ID: ijms-2041151)

All suggestions were entirely accepted and highlighted using the “Track Changes” function in the text.

Please find our comments below.

Sinceriously yours,

Carlos Priminho Pirovani

  1. May I know if Solanum lycopersicum, var. MicroTom's genetic information is archived in a database? If yes, please indicate the database and accession number in the manuscript. If no,please indicate the source of the plant instead in the manuscript. Information has been added to the text better explain the introduction – Line 80 - 81

Reviewer #1

  1. First of all, the selection of this protein is only mentioned in the introduction but actually, there is no further information why it was selected in the result section. That raises questions and it would be valid to request additional reasoning in the result section to select this protein. – Information has been added to the text better explain the methodology, results and discussion – Line 83 – 97; Line 237 – 261 and Line 454 - 466

  1. Then, the protein/gene has been retrieved from a fungus that is causing WBD of the cocoa tree. All the work is performed only on tomato, so what is actually the biological relevance. Has there been any test on cocoa? If so, include the information, if not, why? We appreciate your comment and agree how important this biological question from BASIDIN regarding cacao is. The fungus perniciosa that causes WBD is a pathogen whose main host is T. cacao, however, it is reported in the literature as causing characteristic symptoms of WBD in other species of the cacao family, such as cupuaçu (Theobroma grandiflorum), and in other host species belonging to the Solanaceae, Malpighiaceae, Bignoniaceae and Bixaceae families (Thorold, 1975; Bastos e Evans, 1985; Bastos e Andebrhan, 1986; Griffth, 1989; Luz et al. 1997; Bezerra et al, 1998, Ferraz et al. 2016). The phytopathogen has been reported infecting several cultivated solanaceous plants such as eggplant, sweet pepper, pepper, and tomato, all of high economic importance. These plants when infected show similar symptoms to that described in cocoa, which present exhibit swelling of stem, terminal and axillary brooms, tissue necrosis and presence of hyphae with clamp connections (Bastos e Evans, 1985; Luz et al.1997; Ceita et al. 2007; Garcia et al. 2007). Therefore, the choice of tomato for BASIDIN biological function assays was based on its being a host of M. perniciosa with characteristic symptoms of WBD and mainly because it is a model plant, with fast growth, when compared to T. cacao, despite being the main host of M. perniciosa, due to its worldwide socioeconomic importance, is a slow-growing plant, its tissues, in particular the leaves and meristems, have a very high level mucilaginous polysaccharides and phenolic compounds, as it is a recalcitrant plant (Figueira et al. 1994; Gesteira et al. 2007) that difficult molecular manipulation. These characteristics must be taken into account when designing any biological assay with cocoa.

Recognizing this, for this work we aimed to characterize BASIDIN biochemically and functionally in a model plant, in this case the tomato. However, studies in our research group with another team are already being developed, with the objective of evaluating the biological function of BASIDIN in different hosts of M. perniciosa, including T. cacao, and this was possible thanks to the promising results of BASIDIN in tomato. This team is studding the capacity of BASIDIN to induce mechanisms of defense in host and no host plants. Therefore, with the publication of this study indicating BASIDIN as an effector of M. perniciosa, it will be possible to reference it in future studies that we are developing.

  1. Which type of the fungus (S or C) is used as reference here (only discussed that they exist in line 264, not indicated which) and are there changes (AA) in the protein that might be relevant.- Another major point is the production of the recombinant protein, what was produced exactly, the full protein, a “cleaved” version, with or without signal peptide? Information has been added to the text better explain the methodology – Line 456; Line 472 – 475.

  1. In the figure 1 legend suddenly a Histidine tail is mentioned but no further information is provided in the manuscript. Information has been added to the text better explain the methodology – Line 472 – 475.

  1. The protein of 162 residues indicated in the text (GenBank: EEB89748.1), line 79) should be (with signal peptide) 17.81KD, with the addition of a His(6*), you get at 18.63 kD but in Figure 1, the band is according to legend 20 KD, but in fact is slightly bigger. If you remove the signal peptide (which should be extremely logic for a secreted protein) and add a His(6), the difference gets even bigger (should be only 16.82 kD). This should be addressed in full detail what has been done!! Information has been added to the text better explain results and discussion – Line 83 - 97; Line 237 – 261

  1. There seems some mix-up in concentrations which should be carefully addressed, e.g. in the abstract concentrations are mentioned as “4 μmol uL-1 & 11 μmol uL-1”, in line 101 it suddenly is 4 μmol L-1 (and line 114: 11μmol L-1 so from µL-1 to L-1?). Have been corrected and revised accordingly – Line 481 - 494

  1. In the Materials & Methods section, Basidin is used at 3.75 μL L-1 in 5 mmol L-1 which is quite a shift from molars to liters description, and 11.7 μmolL-1 (Line 427, no space included) - Have been corrected and revised accordingly - Line 481 - 494

  1. With the supplemental Figure 1 (assume this should be the file “Supporting Information 29299313_cluster7.pdf), literally there was no clue what I was looking at.. Please provide an explaining legend if even this is indeed the correct figure – Information has been added in the supplementary figure 1.

  1. General remark, shouldn’t all names be written with Italics? Noticed that they are not throughout the whole manuscript. Have been corrected and revised accordingly

Other points raised:

  1. Line 21, if stating for one treatment (spray) a time interval, then also provide a time interval for the wilting effect in the abstract. – Information has been added to the text better explain abstract.

  1. Line 25, should read: that it has - Have been corrected and revised accordingly.

  1. Line 27, “in different species” this is extremely unclear. are we talking about the fungi or plants. Information has been added to the text better explain results and discussion.

  1. Line 27, expressed & important, the importance is not directly linked to levels of expression in my opinion. Have been corrected and revised accordingly.

  1. Line 27, should read: that this protein – Have been corrected and revised accordingly.

  1. Line 50, should read: to unravel the mechanism – Have been corrected and revised accordingly.

  1. Line 104, why was the experiment not continued for a longer time and in addition, the pictures in Figure 2 are quite difficult to interpret (e.g. the zoom in’s) – The results referring to Figure 2 were obtained from detached leaves of Solanum lycopersicum. If the experiment lasted more than 24 hours, the results would be skewed, due to the natural loss of water. This would make it impossible to infer whether the observed symptoms were due to rBASIDIN activity or water loss. Therefore, this part of the work is designed to last up to 24 hours. Information has been added to Figure 2 to facilitate interpretation

  1. Line 123, please add in the text that you stain on H2O2 (title only tells it is produced, no information what is actually done) – Information has been added to the text better explain results.

  1. Line 126, please mention that you spray with the protein, you could spray with anything at a concentration of 4 umol L-1.- Information has been added to the text better explain results.

  1. Line 135 and several other occasion, why is BASIDIN suddenly without capitals. Are we still talking about rBASIDIN? – The manuscript presents two variations of the term: BASIDIN and rBASIDIN. When we refer to recombinant protein that applies from expression in coli to biological assays, the term we use is rBASIDIN. Only for phylogenetic, in silico and transcript that do not include recombinant BASIDIN, we use the term BASIDIN to refer to the study protein. However, we have corrected any inconsistency of terms in the text.

  1. Line 146-147, please provide here or in the discussion a reason for the initial increase after 30 min – Information has been added to the text better explain discussion.

  1. Line 208, double point at the end of sentence - Have been corrected and revised accordingly.

  1. Line 236, what is “it” referring to? To Nep or BASIDIN – Information has been added for clarity.

  1. Line 242 & general remark, we had the results, in the discussion one should not refer to figures. - Have been corrected and revised accordingly.

  1. Line 254, was can? – Have been corrected and revised accordingly.

  1. Line 280, Besides the Greek mythology that is known, there is no other mentioning in the manuscript on EROS? - Have been corrected and revised accordingly.

  1. Line 353, Start of paragraph with “this group”, is difficult to address what is meant. - Have been corrected and revised accordingly.

  1. Line 357, wondering if hypothetical proteins or protein should be used here... I guess that these proteins are deposited as individuals so as protein – We appreciate your inquiry, however, we use the term “hypothetical proteins” because through the results of the BLASTp analysis, in the search for BASIDIN homologues, in the NCBI public database these proteins were identified as hypothetical proteins. Possibly these protein sequences were deduced from the genomic data and have not yet been characterized for biological activity. Therefore, we prefer to use the term that returned from the BLASTp analysis.

  1. Line 401, “suggest that BASIDIN as a novel effector” misses a word - Have been corrected and revised accordingly.

  1. Line 407, elaborate on “synthetic ORF”, meant as produced rather than cloned, or also optimized for E. coli? (if so, provide the optimized sequence) – Information has been added to the methodology for better understanding of the text and the optimized sequence has been made available as supplementary material (Supplementary Figure 3) – Line 472 - 4756

  1. Line 467, why is the degree sign underlined? - Have been corrected and revised accordingly.

  1. Line 479, new protein? rBasidina. – Have been corrected and revised accordingly.

  1. Line 481, what is selected meaning here? Was there a selection made among proteins that show homology or is this just proteins with homology were derived by blasting the NCBI databast using Blast. which blast was used, tBLASTn or BlastP only, please define!! - Information has been added to the methodology for better understanding. The terms have been corrected.

Reviewer 2 Report

The manuscript entitled "BASIDIN as a new protein effector of the phytopathogen causing witch’s broom disease in cocoa" investigated the possible function of BASIDIN, a new protein effector, through bioassay with model tomato plants using recombinant protein expressed in a heterologous system, results showed that rBASIDIN treatment caused wilting symptoms, hydrogen peroxide production and leaf membrane destroy. Phylogenetic analysis further indicated that BASIDIN is related to proteins from basidiomycete fungi. The data presented in this manuscript suggest that BASIDIN as a novel effector of the fungus M. perniciosa. But some methodologies, results and discussion are not scientific and reasonable.

Major comment

1. The concentration of rBASIDIN used in this manuscript were not consistent, including 4 μmol L-1 for tomato leaf injection, 11µmol L-1 for tomato leaf spraying, 10 μmol L-1 for gas exchange analysis, 4 μmol L-1 for electrolyte leakage analysis, please provide the basis for concentration selection.

2. Why select rMpNEP2 as positive control?

3. Both rBASIDIN injection and spraying on tomato leaves caused wilting symptoms, are the mechanisms of the two treatments causing the same symptoms consistent? Please explain and discussion?

4. Line 401-402 Authors suggest that BASIDIN as a novel effector of the fungus M.perniciosa, which contributes to activation of plant innate immunity. The data presented in this article do not support this conclusion, especially “contributes to activation of plant innate immunity”.

5. English should be improved; grammar needs enhancement in many sentences and paragraphs.

Minor comment

1. Line 24-25 the sentence Phylogenetic analysis indicated that has orthologs in other basidiomycetes that also cause diseases in plants of economic importance” is unclear, please re-write.

2. All pathogen names should be italic

Author Response

We sincerely appreciate yours contribution in the review and suggestions regarding the manuscript "BASIDIN as a new protein effector of the phytopathogen causing
witch’s broom disease in cocoa
." (Manuscript ID: ijms-2041151)

All suggestions were entirely accepted and highlighted using the “Track Changes” function in the text.

Please find our comments below.

Sinceriously yours,

Carlos Priminho Pirovani

  1. The concentration of rBASIDIN used in this manuscript were not consistent, including 4 μmol L-1 for tomato leaf injection, 11µmol L-1 for tomato leaf spraying, 10 μmol L-1 for gas exchange analysis, 4 μmol L-1 forelectrolyte leakage analysis, please provide the basis for concentration selection – This comment is very pertinent. However, in the protein production phase in a heterologous system, we had some difficulties in obtaining concentrated rBASIDIN. Because it is a meticulous technique, depending on the working protein, the expression may vary, consequently varying the concentration obtained. Therefore, we had to work with different concentrations of rBASIDIN. However, for each assay we performed tests of concentrations based on the literature. In the process of injection in tomato leaves, the concentrations were based on Garcia et al. 2007. In the spraying stage, preliminary tests were carried out with concentrations ranging from 0.01µmol L-1 to 11 µmol L-1, with 10µmol L-1 being the concentrations that obtained the best results. For gas exchange analysis, the choice of 10µmol L-1 was made based on what we observed in the initial hours in the spray tests shown in supplementary figure 3. Since in this test 11µmol L-1 caused damage to the leaves, we opted to lower the concentration to 10µmol L-1 in the gas exchange analysis to prevent the leaf from being damaged and unable to perform gas exchange measurements. For the analysis of electrolyte leakage based on previous tests (spraying) we chose to select two concentrations 4µmol L-1 and 10µmol L-1. Information has been added to the methodology to better explain the concentrations used.

  1. Why select rMpNEP2 as positive control? – We chose rMpNEP2 as a positive control due to the availability of the protein expression clone in our lab. We also consider the fact that this effector has already been tested and causes symptoms of necrosis (Garcia et al. 2007; Teixiera et al. 2014; Villela-Dias et al 2014)

  1. Both rBASIDIN injection and spraying on tomato leaves caused wilting symptoms, are the mechanisms of the two treatments causing the same symptoms consistent? Please explain and discussion? We appreciate your questioning and reflection on the data. Although the bioassays were carried out by microinjection and spraying, the results are consistent and complementary. In the microinjection method, the leaves were detached from the plant. rBASIDIN was inoculated at a concentration of 4µmol L-1 in 20 µl, observed in a time interval of 0 – 24 hours. Within 6 h wilting symptoms were observed. The wilting symptom in the spraying bioassay was previously observed at a time of 2h. Even taking into account that in the spraying bioassay rBASIDIN was at a higher concentration (11µmol L-1), the results are consistent because wilting was previously detected, disregarding that the microinjection result could be related to the natural wilting process of the leaf, as this was highlighted. Furthermore, the same amount of solution with rBASIDIN at a concentration of 11µmol L-1 was boiled at 80ºC and injected. After observation, wilting symptoms were also detected within 6 hours, which validates the microinjection and spraying bioassay, confirming that the wilting symptoms were caused by the protein and not by any inconsistency of the mechanisms, which despite the protein being deposited in different locations (mesophyll and leaf surface) the symptoms were detected in the entire tissue and not in the regions of action of each mechanism (Figure 2, Figure 3 and Figure 6). In microinjection, the application of the protein initially reaches a smaller leaf area, while in spraying, a larger area of ​​the leaf surface is reached.

  1. Line 401-402 Authors suggest that BASIDIN as a novel effector of the fungus perniciosa, which contributes to activation of plant innate immunity. The data presented in this article do not support this conclusion, especially “contributes to activation of plant innate immunity”. – Have been corrected and revised accordingly – Line 445 - 452

  1. English should be improved; grammar needs enhancement in many sentences and paragraphs - The entire document was checked by an English native speaker.

Minor comment

  1. Line 24-25 the sentence “Phylogenetic analysis indicated that has orthologs in other basidiomycetes that also cause diseases in plants of economic importance” is unclear, please re-write Have been corrected and revised accordingly.
  2. All pathogen names should be italic - Have been corrected and revised accordingly.

Reviewer 3 Report

Keilane Silva Farias and colleagues present an article on the initial characterization of a secreted effector protein in Moniliophthora perniciosa, that appears to be involved in hydrogen peroxide production, damaging the leaves and affecting the photosynthetic rate. In particular, this putative effector was purified after heterologous expression and its effects on plant leaves were quantified by using bioassays and estimation of H2O2 production, photosynthetic rate and stomatal conductance. In addition, a secondary structure was identified by circular dichroism and potential evolutionary relationships were discussed along with expression analyses from existing seq library data.

In general, the article contains some interesting and potentially important information for people working in this field, however, what seems to be essentially missing is the proof that this protein is indeed secreted and also expressed during infection, aspects that are crucial to fully support the conclusions and highlight novelty.

Author Response

We sincerely appreciate yours contribution in the review and suggestions regarding the manuscript "BASIDIN as a new protein effector of the phytopathogen causing
witch’s broom disease in cocoa
." (Manuscript ID: ijms-2041151)

All suggestions were entirely accepted and highlighted using the “Track Changes” function in the text.

Please find our comments below.

Sinceriously yours,

Carlos Priminho Pirovani

  1. In general, the article contains some interesting and potentially important information for people working in this field, however, what seems to be essentially missing is the proof that this protein is indeed secreted and also expressed during infection, aspects that are crucial to fully support the conclusions and highlight novelty. We appreciate your comment and confirm that this is being discussed and planned in our research group, as the next objectives of studies with BASIDIN. Our team is made master's and doctoral student and we have a new team engaged in designing an experiment with comparative proteomics to identify whether BASIDIN is secreted and what is its expression level in the pathosystem perniciosaT. cacao. These analyzes will be done via LC/MS-MS gel free and by Western blot and qPCR in real time. In the next publications we intend to answer all possible questions about BASIDIN's action and move on to other steps, such as the development of biotechnological products. Although this manuscript does not answer this question, it sheds light on the biochemical and functional characterization of BASIDIN, presenting important results for the area, achieving the objective of the study, which was to characterize a potential effector protein, selected in our genomic bank under the code Mp4145 -3305 that has 100% identity with the predict protein (GenBank: EEB89748.1) in the first genomic sequencing of the fungus M. perniciosa (Mondego et al. 2008), since many potential effectors were predicted in M. perniciosa, this being the first characterized effector of the Barbosa et al. 2018.

Language review - The entire document was checked by an English native speaker.

Round 2

Reviewer 1 Report

Many thanks for addressing the comments which in most cases clarified the manuscript. Nevertheless, some issues were incorporated that need to be addressed carefully. Especially the missing supplemental files and the swapping of the ones present is clearly disturbing...

-line 22, a space is missing before the concentration indicated
-line 246, please update reference, they should be numerical
-line 256 and other places, some references are in italic others not..

-line 455, spelling of Basidin?

-Line 456, modify reference to numerical

-Line 487, please correct reference

-Line 553, keep one way of writing Blastp  (see line 460..)

-Line 583, writing of BASIDIN? Also in legend of provided Supplemental Figure 1

-Line 588, Supplementary Figure 3 is showing leaves, not a sequence. In addition, if the authors state that the codon usage was optimized but not the amino acid sequence (Lines 472-474), then what is the reason to show a protein sequence? Please provide the optimized codon sequence (with or without protein seq).

Line 591, I guess Supplemental Figure 4 is the now actually given Supplemental Figure 3, please also provide information of plant species shown.

Line 599, Supp. Fig. 5 is not in the files provided.

In my downloaded supplemental files, the Supplemented Tables are swapped by their number (1=2 and 2=1), Textual mistakes in the file, e.g. tabe instead of table.

In Supplementary Table 1 (now 2) is referred to Figure 7, but not all fungi in Figure 7 are mentioned in the table.

Careful reading of the whole manuscript is still recommended.

Reviewer 2 Report

All my suggestions Authors took into account. Now I recommend the article for publication.

Reviewer 3 Report

I understand and appreciate the authors’ point, however, the provided information is still not fully supporting the conclusions drawn, and therefore, in my opinion, the manuscript is not appropriate for a journal like IJMS.